

# Quantifying Vertical Hyporheic Exchange and hyporheic residence time in thalweg paths of meandering streams characterized by multiple riffle-pool sequences morphology

Aminreza Meghdadi[1*], Morteza Eyvazi[2], Zohre Najatijahromi[3], Bahram Saghafian[4]

[1] PAYARAH Consultant Engineering Company, Department of Research and Engineering, Zanjan, Iran. Email: Meghdadi.aminreza@gmail.com ([*]Corresponding Author)

[2] Department of Agriculture and Plant Breeding, Faculty of Agriculture, Zanjan University, Zanjan, Iran. Email: m.eyvazi@znu.ac.ir

[3] Faculty of Earth Sciences, Department of Minerals and Hydrogeology, Shahid Beheshti

University, Tehran, Iran. Email: zo_jahromi@sbu.ac.ir

[4] Civil Engineering Department, Islamic Azad University, Science and Research Branch, Tehran, Iran. Email: b.saghafian@srbiau.ac.ir

**Abstract**

Riffle-pool sequences in the thalweg paths of meandering streams are of pivotal importance to

the hyporheic exchange pattern in a fluvial network, but the complex hydrodynamic, morphological, and sedimentary features of riverbed sediments increase the difficulties associated with vertical hyporheic exchange (VHE) quantification. This study applied depth-dependent radon ($^{222}$Rn) and diel temperature variations to quantify VHE and residence time ($t_r$). The study was conducted in four different hyporheic areas with riffle-pool sequences in

the third-order Ghezel-Ozan River, located in north-west Iran. The mean values of temperature-derived VHE ($VHE_T$) and radon-derived VHE ($VHE_{Rn}$) were 0.67±0.32 m/day and 0.63±0.36 m/day, respectively. Due to effects of sediment bed heterogeneity on temperature variation and $^{222}$Rn activity at downwelling and upwelling points, there were discrepancies between radon-



derived ($tr_{Rn}$) and temperature-derived residence time ($tr_T$), with mean values of $2.11\pm1.17$ days and $1.87\pm1.26$ days, respectively. The value of $tr_T$ was well within uncertainty boundaries at a 95 percent confidence interval ($p<0.05$) and was lower than $tr_{Rn}$ at the downwelling points. The analysis of vertical diel temperature, radon and electrical conductivity variations revealed

subsurface water exchange to be greatly affected by larger scale regional flow. The comparison between $VHE_T$ and $VHE_{Rn}$ with VHE obtained from PHAST model simulation ($VHE_{PHAST}$) revealed a higher correlation between $VHE_T$ and $VHE_{PHAST}$ ($R^2=0.96$) than with $VHE_{Rn}$ ($R^2=0.76$). Furthermore, vertical hydraulic conductivity ($K_v$) of the sediment-bed materials, calculated in situ by the permeameter test, indicated not only that $K_v$ was up to 21% higher in

areas dominated by upward movement than at downwelling points, but also principle component analysis (PCA) demonstrated the dependence of Kv on porosity, VHE, and %sand of the stream-bed materials. This study provides evidence that vertical flux in the hyporheic zone is mainly affected by stream sinuosity and regional subsurface flow, and that the temperature method is more suitable than radon activity to quantify hyporheic exchange

patterns.

**Key words:** vertical hyporheic exchange, [222]Rn, temperature, residence time, PHAST simulator, Ghezel-Ozan River

## 1. Introduction

Infiltration of stream water into saturated sediments beneath stream bed, and then exfiltration

into the stream after intra-sediment residence time (Gooseff, 2010; Tonina, 2012), is recognized as hyporheic exchange. The hyporheic zone, consisting of the saturated and kinematic zones beneath and adjacent to the stream bed which connect to the river aquifer system, is known as a key area for regulation of the dynamic biogeochemical properties of the exchange water (Deng et al., 2015). Therefore, accurate quantification and identification of the





spatial patterns of the hyporheic exchange process plays a crucial role in determining the fate and transport of anthropogenic contaminants (Meghdadi and Javar, 2018).

Hyporheic exchange is mainly governed by riverbed morphological features such as riffle-pool sequences (Gariglio et al., 2013), stream sinuosity and curvature (Meghdadi and Eyvazi, 2017; Zhang et al., 2017), and stream sedimentary properties (Song et al., 2017). These complex and varied stream morphological features increase the difficulties in understanding the magnitude of the hyporheic exchange.

Riffle-pool sequences in the thalweg areas of a meandering river are characterised by complex morpho-dynamic (Bätz et al., 2016) and hydrodynamic features, which create a sequence of stagnation and accretion zones in the direction of river flow. As a result, the magnitude of the hyporheic exchange is highly affected by the morphological variations (Lambs, 2004). Hence, understanding the effect of the different hydrodynamic and morphological variations on the vertical hyporheic flux patterns plays a pivotal role in appropriate regulatory decision making concerning contaminants (Criswell, 2016). The stream sediment vertical hydraulic conductivity is another key parameter for analysis of stream-aquifer connectivity and, due to variation in river morphology, is closely linked to the hyporheic exchange (Stewardson et al., 2016).

Environmental natural tracers have been widely proposed to assess the vertical hyporheic exchange (VHE) in river-aquifer systems. Three environmental tracers that have been applied for this purpose are temperature, electrical conductivity (EC), and radon ($^{222}$Rn). Comparison of the magnitude of diel temperature variations in stream and subsurface sediments, at one or more depths, provides a useful insight into the magnitude and direction of hyporheic water flux. Typically, in upwelling location subsurface sediments the diel temperature is lower, whereas a rapid increase in subsurface diel temperature results from downward hyporheic flux.





Radon, which is a short-lived radioactive gas, is employed as a tracer for up to medium-term residence time (<15 days) when flux assessment is of particular interest (Atkins et al., 2016). River water infiltrate into subsurface sediments brings about a rapid increase of the radon activity of the infiltrated water with time, and allows the water characteristics to be measured 5 (Cook et al., 2011). EC in hyporheic zones where the groundwater endmembers have distinct signatures can be employed to distinguish between stream water and regional subsurface flow (McCallum et al., 2010).

Radon and temperature have been widely suggested for quantification of VHE, but there has been limited research to investigate their reliability in accurately estimating hyporheic 10 residence time and hyporheic flux. Choosing between radon and temperature is a key challenge for precise estimation of hyporheic exchange. For example, Rau et al. (2012) experimentally determined that the temperature-derived flux in homogeneous saturated sediments was up to 20% higher than the radon-derived flux, with this difference arising from inhomogeneous heat transport through the sediments. In another study, Schornberg et al. (2010) demonstrated that 15 the disparity between temperature-derived and radon-derived flux dramatically increased with increased variation in sediment hydraulic conductivity.

Recently, Gooseff et al. (2009) performed a study to model upwelling and downwelling locations in hyporheic zone of mountain streams using longitudinal channel unit spacing profile. In other study, Wondzell et al. (2009) tested groundwater flow models to evaluate the 20 amount of residence time and exchange flow in hyporheic zone of a mountain stream and they showed that a model accuracy strongly influenced by the choice of nodal spacing as well as procedures applied to interpolate spatially distributed parameters to the model domain. Furthermore, Constantz (2008) reviewed the application of heat as a tracer to assess shallow ground water movement and described the recent temperature-base approached to evaluate





hyporheic exchange. The storage capacity of hyporheic sediment was assessed recently by Neilson et al. (2010) via simultaneous application of temperature and solute model to better calculate transient storage modelling approaches.

Modelling tools provide another indispensable approach to visualise the spatial distribution of dynamic hyporheic vertical flux. Recently, the PHAST model, developed by integration of two previously studied simulation models PHREEQC and HST3D, was used to model a wide range of kinetic and geochemical reactions (Parkhurst et al., 2010). PHAST is capable of quantitative visualisation of the downward and upward flux locations of stream beds. In brief, it employs a set of partial differential equations for solute transport and a set of nonlinear algebraic differential equations for solute chemistry. The saturated groundwater flow equation is applied in the model to estimate total subsurface fluid mass (Parkhurst et al., 2010). The subsurface solute transport and flux equations are integrated via the dependence of advective-dispersive transport on the interstitial flux-velocity field. By applying a successive solution approach for flux reaction and transport calculations, numerical results are obtained for each of the dependent variables including, solute and species concentration, potentiometric head, and the mass of reactants in each grid cell.

Hence, providing an innovative alternative framework incorporating both morphological and physio-chemical characteristics of streams which can be quantitatively applied for more accurate spatial evaluations and modelling of the hyporheic exchange within thalweg paths of sinusal streams is area of active research. Therefore, the aims of this study were to: (a) characterise the hydraulic and morphological features of the thalweg paths of meandering streams and understand the spatial distribution of hydraulic conductivity; (b) to quantify vertical hyporheic exchange in thalweg paths characterised by riffle-pool sequences; (c) apply the vertical variations of diel temperature, radon, and EC to determine the depth-dependent



variability of subsurface flux; and (d) investigate the suitability of PHAST modelling technique to estimate VHE and associated residence times, and verify the accuracy of the modelling results against radon and temperature.

## 2. Sampling and study area description

### 2.1 Study area

This study was conducted in the Tarom watershed, which is a sub-basin of the Ghezel-Ozan watershed. The Tarom watershed is located at 37°01'48" to 36°54'36" latitude and 48°44'24" to 48°54'24" longitude, covering a total area of 113 km$^2$ (Figure 1A). The Ghezel-Ozan River passes through the Tarom valley and is approximately 80 km north-west of Zanjan city. The river is a highly meandering stream and consists of multiple riffle-pool sequences and sandy clay alluvial sediment textures. Igneous bedrock is the dominant geochemical unit underlying the river sediments. Furthermore, the geology of the study watershed is mainly late Tertiary to early Quaternary calcareous sands, basalt, and clay (Meghdadi and Eyvazi, 2017). Four zones located in the thalweg path of the sinuously flowing stream and characterised by different morphological and hydrodynamic features (multiple riffle-pool sequences, Figure 1A) were chosen to fulfil the study requirements.

### 2.2 Sampling and field data measurements

Field data sampling was carried out along more than 1600 m of the Ghezel-Ozan River. Due to the suggestion by Anibas et al. (2011, 2009) and Gariglio et al. (2013) that winter is the most favourable time for assessment of the thermal process, the sampling procedure was conducted from 02 January 2019 to 16 January 2019 across the four separate zones. In total there were 33 testing points and 11 cross sections arranged perpendicularly or parallel to the stream flow direction (Figure 1B). A set of 33 multi-level sampler (MLS) piezometers (Meghdadi and Eyvazi, 2017) was constructed using galvanized steel pipe with inside diameter (ID) of 70 mm





and an end-fitted steel drive point that was vertically driven into the river sediments at each testing point using a sledge hammer. Five Solinst TLC level loggers were placed into each MLS and fastened with a steel cable to the threaded lid (Figure 1B). The level loggers were calibrated before and after each measurement to ±0.5 °C accuracy. The TLC level loggers were

set at 0.25, 0.45, 0.65, 0.85 and 1.05 m depths and recorded the temperature at each depth at 30 minute time intervals, which is the time interval required to satisfy the requirements of the sediment thermal calculation under unsteady state flow conditions  (Boano et al., 2013).

Besides the data loggers, at each of the five depths, flexible polytetrafluoroethylene (PTFE) water sampling tubes with 5 mm ID connected to a peristaltic pump were employed for pore

water sampling (Figure 1B). Furthermore, a 1.5 m long high-density polyethylene tube with 70 mm ID and fitted with a stainless steel drive point, for ease of penetration into sediments, was employed to measure the hydraulic head (Figure 1B). The piezometer had a 10 mm aperture located 12 mm above the stainless steel joint and was used to measure the hydraulic head once at each testing point.

Three sediment cores from each zone were collected, making 12 cores in total, using 3" Shelby tube samplers (Forsum Ultra-hard Material Industry Co. Ltd, China) to minimise sample disturbance and calculate soil-related parameters such as porosity, $D_{50}$ (mid-point range of the particle size distribution), bulk density, and pore water radon equilibrium activity. To measure the radon activity, surface water samples at 0.05 m depth, and sediment pore water samples

from the five aforementioned depths at each of the 33 testing points, were collected using MLS and stored in 40 mL glass bottles, then the radon activity was calculated in the field using RAD&-H$_2$O (Durridge Co, MA, USA). The RAD&-H$_2$O were formerly calibrated to a precision of approximately 1%. To measure the radon equilibrium activity of the aquifer sediments, soil samples were collected in 250 mL pre-rinsed borosilicate glass bottles then

filled completely with distilled water and sealed so that no air remained in the bottle. These





were left in the laboratory at 20 °C for 1.5 months to reach the equilibrium concentration. Triplicate water samples were sent to the geochemical laboratory at the University of Zanjan for equilibrium concentration analysis using the liquid scintillation method (Leaney and Herczeg, 2006). A hand held YSI probe (±1% accuracy) was employed to measure the EC in

the field. All the sampling and in situ data measurement procedures described above were carried out based on the criteria described by APHA (2005). The quality control and quality assurance (QC/QA) procedures for all the above-mentioned in situ analyses were achieved using three replicas of each experimental phase and analysis of one surrogated blank per 25 samples. Results from analysis of the replicas were within ±5% of the standard deviation

(Hounslow, 1995).

## 3. Methodology

### 3.1 Temperature

Diel temperature variation in the hyporheic zone and surface water bodies provides a useful insight for estimation of the magnitude of VHE between the two water bodies. In this study,

the conductive-advective heat transport procedure employing the one-dimensional advective-diffusion equation was applied to quantify the groundwater/surface water exchange rate and evaluate the hyporheic vertical flow pattern (Hyun et al., 2011). The equation for one-dimensional advection-diffusion heat transport can be expressed as equation 1 (Hatch et al., 2006; Naranjo et al., 2013):

$$\frac{\partial T}{\partial t} = k_e \frac{\partial^2 T}{\partial z^2} - \frac{q}{\gamma} \frac{\partial T}{\partial z} \qquad (1)$$

where T is stream-bed temperature (°C), t is time (s), z is vertical depth in the stream-bed (m), $k_e$ is the effective thermal diffusivity (m²/day), q is the Darcy vertical pore water flux ascribed to pore water velocity (v in ms⁻¹, v=q/θ, where θ is the porosity of the saturated sediments), and



$\gamma$ is the ratio of the volumetric heat capacity of saturated sediments ($\rho C$) to the heat capacity of water ($\rho_f C_f = 4.22 \times 10^6$ J/(m$^3$K)); the magnitude of $\rho C$ ($2.71 \times 10^6$ J/(m$^3$K)) is derived based on the procedures described by Waples and Waples (2004). The values of $\gamma$ and $k_e$ applied in this study were obtained from the experimental relationships established in the literature (Lapham,

1989; Lunardini, 1981) for sandy-clay soil ($k_e = 2.44 \times 10^6$ J s$^{-1}$m$^{-1}$K$^{-1}$). The diel temperature variation of the river were approximated with a sinusal function and the Equation 1 was analytically solved to calculate the vertical water flux. In the Equation 1, the upper boundary corresponds to the stream bed surface (z=0 where TZ=T0) and the lowermost sampling depth (Z=L, L=1.05 m) sets the lowest boundary (Tz=TL at z=L). ). Considering the boundary

condition, equation 1 can be expressed as equation 2:

$$\frac{T - T_0}{T_L - T_0} = \frac{e^{\beta z/L} - 1}{e^{\beta} - 1} \tag{2}$$

where $\beta$ is equal to $\rho_f C_f qL/k_e$ (dimensionless parameter), which is negative for upwelling flux and positive for downwelling flux. As described by Arriaga and Leap (2004), the value of q is proportionate to the $\beta$ value: when $\beta$ is positive, q is positive. The temperature values at 20

locations in the sediment layers were used to calculate the $\beta$ value and subsequently solve equation 2. The diel temperatures at each of the 33 testing points were simulated independently using parameter estimation software PEST (Doherty, 2010), which showed the temperature profile over each 12 hour period of the total 24 hour recording time. During the PEST analysis the sum of the square error between the observed and simulated temperature data was

minimized via repetitive variation of q until the line of best fit was achieved. The VHE derived from temperature values (VHE$_T$, m/day) were obtained by solving equation 2 using the Microsoft Office Excel solver add-in to calculate $\beta$, then based on the method described by Arriaga and Leap (2006), as equation 3 the VHE can be achieved:

$$\left| VHE \right| = \frac{K_e \beta}{C_f \rho_f L} \tag{3}$$





The magnitude of the temperature-derived residence time ($tr_T$) at each depth (z) can be

calculated as $t_{rT} = \frac{z\theta}{VHE_T}$. The upper and lower uncertainty ranges of $tr_T$ were determined by

the Monte Carlo analysis (parameter optimisation method), considering over 580 simulations

of temperature transport values and different $k_e$ and VHET magnitudes. Random $k_e$ values were

generated by assessing a uniform distribution of the estimated $k_e$ up to a variation of ±50%.

### 3.2 Vertical hydraulic conductivity

Vertical hydraulic conductivity ($K_v$) is an important hydraulic parameter for estimating the

interaction between groundwater and surface water (Jiang et al., 2015). In this study, the

magnitude of the stream bed hydraulic conductivity was calculated using equation 4 from

measurements at each of the 33 testing points (all the calculation process were performed based

on the procedures described by Song et al. (2017).

$$K_v = \frac{L_v}{t_2 - t_1} \times Ln(\frac{h_1}{h_2}) \tag{4}$$

In equation 4, $h_1$ and $h_2$ are the hydraulic heads measured at time t1 and t2, respectively, and

$L_v$ is the stream sediment thickness in metres (Figure 1B). The length of $L_v$ was approximately

1.05 m and the average ratio of $K_v$ to ID was 7.9. As an assessment of the accuracy and

applicability of the equation 4 method, the calculated values of the potential errors for the $K_v$

measurement were less than 5%. Hence, the associated in situ measured parameter (during the

rest period) could be employed to assess the magnitude of $K_v$. Furthermore, to identify the most

influential parameters on $K_v$, orthogonal projection and transformation analysis of soil physical

and hydrodynamic parameters (%sand, %silt, %clay, $t_r$, $D_{50}$, flux velocity, porosity, VHE, ſ,

and stream depth) was performed using principle component analysis.

### 3.3 Radon





Noble gas radon ($^{222}$Rn), with a half-life of 3.82 days, is found both in streams and aquifers. $^{222}$Rn is produced by radium ($^{226}$Ra) decay: dissolved radium can attach to the aquifer matrix under low salinity conditions (Cook and Herczeg, 2012) and the dissolved radon concentration in groundwater increases over time due to $^{222}$Rn production in the aquifer sediments until an

equilibrium is reached. However, the magnitude of the equilibrium concentration measured from the sampled sediments was dependant on the radon activity rate of the aquifer material, as approximately four weeks is required for complete equilibrium to be attained. Theoretically, the amount of water that entered the aquifer within this time can be measured. In this study, the radon activity at time t (equation 5) was applied to estimate the radon-derived VHE rate

(VHE$_{Rn}$, m/day), as follows (Cecil and Green, 2000):

$$A_t = A_e\left(1 - e^{-\lambda t_{rRn}}\right) + A_0 e^{-\lambda t\_rRn} \tag{5}$$

In equation 5, A$_t$, A$_e$, and A$_0$ refer to the radon activity at time t, equilibrium, and initial time of sampling (t=0), respectively. $\lambda$ is the $^{222}$Rn decay coefficient which is 0.181/day, and t$_{rRn}$ is the radon-derived residence time which can be achieved by solving equation 5. The uncertainty

range of t$_{rRn}$ was evaluated for each of the 33 testing points employing the Monte Carlo uncertainty analysis procedure (at 5[th] and 95[th] percentiles). The mean and standard deviation values of the radon activity of stream water, and at equilibrium in the sediment pore waters, were applied to generate the normal distribution pattern of A$_0$ and A$_e$. About 3590 simulations of the Monte Carlo analysis considering A$_e$ and A$_0$ were used to define the upper and the lower

error boundaries of t$_{rRn}$. The estimated t$_{rRn}$ was converted to the radon-derived VHE according to VHE$_{Rn}$=z$\theta$/t$_{rRn}$.

**3.4 PHAST modelling**



The computer simulation program PHAST (PHREEQC and HST3D) is a versatile model which can simulate solute transport in two- and three-dimensional (2D/3D) saturated groundwater systems (Parkhurst et al., 2010). The calculation of flow transport is based on the modified HST3D version. The major benefit of PHAST is to allow input of the 2D spatial distribution

of groundwater flux using a combined map of the grid coordinate system rather than node by node data insertion. Different boundary conditions including specified flux, specified head difference, and head-dependent flux, are available in the software. Furthermore, output data (HDF files) can be visualized in 2D format using the ModelMuse graphical interface (Bushira et al., 2017).

In this study, to simulate vertical hyporheic flow the model values for porous media, grids, initial and boundary conditions, solute transport, and time intervals, were defined in the model based on the data file provided by Zanjan's Natural Resources Office Division of Water Resources Management (http://www.frw.org.ir/00/En/default.aspx). The initial water table in PHAST applied the 2D data set that is only defined in the model for one layer of nodes.

The river boundary condition (RIVER data block) was used to simulate the river aquifer water exchange which uses the sign of head difference between aquifer and the river, the layer thickness, and the hydraulic conductivity to assess the water exchange between two water bodies. For this study, the river boundary condition was transformed into source terms on the cell by cell basis which the discretised equation for a river segment (s) and for the water table

cell (wt) is described as equation 6, as follow:

$$Q_{Rs} = \frac{K_{Rs}}{b_{Rs}}(h_{Rs} - h_{wt}^n)S_{Rs} - \frac{K_{Rs}}{b_{Rs}}S_{Rs}\delta h_{wt} \qquad (6)$$

Where $Q_{Rs}$ is the volumetric flowrate for river segment (m3/s), $b_{Rs}$ is the riverbed thickness for the segment s (m), $K_{Rs}$ riverbed hydraulic conductivity for the segment s (m/s), $h_{Rs}$ is potentiometric head for segment s (m), n refers to the discrete of the time value during the





simulation, $S_{Rs}$ denotes the area of the river segment (m$^2$), and $\delta_{hwt}$ refers to the variation of the water-table elevation over the time (m).

## 4. Results and discussion

### 4.1 Morphological characteristics of the study zones

Detailed streambed elevation data from the four study zones is illustrated in Figure 2. In each of the four zones the clear riffle-pool sequences, as well as the erosional zone in the vicinity of the thalweg path, and the depositional zone, could be observed. The different bed-form structure in association with the distribution of sediment grain size was used to characterise the depositional and erosional areas, with the erosional zones mainly observed at the left bank side

of the parafluvial area toward the thalweg paths.

In addition, based on the river bed morphology and sediment characteristics, 33 testing points along the riffle-pool sequence of the meandering river were identified. Based on the platform geometry and flux momentum, these sampling points were clearly interconnected (Hiatt and Passalacqua, 2015). The study zones were in a region of river sinuosity, and contained several

riffle-pool sequences with flux momentum of <0.5 and acute angle, causing stagnation and decreased flow velocity (Riley et al., 2015); the flux momentum ratio ($M_r$<0.5) and sharp river curvature leads to anomalies in river bed sediment temperature especially during high flow conditions (Riley and Rhoads, 2012; Xian et al., 2017; Zhang et al., 2017). In contrast, downstream of the river deflection is characterised as a flow accelerator and recovery zone.

The sequences of flow accelerator and diffraction areas, and their integration with riffle-pool sequences, provides a complex river-aquifer interaction mechanism. The area selected for this study was in accordance with the approach of Johnson (2015), where the succession of river accretion and deflection zones played a significant role in identification of subsurface flux patterns and quantification of the magnitude of VHE.





Grain size distribution and median grain size and porosity had a clear influence on the vertical hydraulic conductivity $K_v$ and VHE. The parameters relating to sediment grain size distribution in zones 1-4 are provided in Table 1 from which it can be seen that there was a high cumulative percentage of grain sizes between 0.075 mm and 2 mm observed in each of the zones.

## 4.2 Assessing the vertical hydraulic conductivity of sediments

The values of $K_v$ ranged from 0.27 to 3.76 m/day with a coefficient of variation (CV) of 0.45 to 0.69. These low CV values are typically ascribed to the sandy clay soil texture of the river sediment which exerts an important controlling factor on $K_v$ (Chen et al., 2013; Jorda et al., 2015). The orthogonal transformation and projection of the physical and hydrodynamic related parameters on $K_v$ were described using the principle component analysis (PCA) method (Donath et al. (2015); Figure 3).

The PCA results accounted for more than 65% of the total variance (PC1=48.3% and PC2=17.2%). There was significant correlation between temperature-derived $K_v$ and VHE (see section 1) and porosity, as well as with percentage of sand ($p<0.05$), which suggested that it was the properties of the river-sediment grain that had the highest impact on $K_v$. The second principle component (PC2) accounted for lower variance, but the high value of PC2 for $D_{50}$ ($p<0.05$) could indicate the controlling effect of the magnitude of the median grain size on $K_v$. The negative value of the first principal component for temperature-derived residence time with $K_T$ ($p<0.05$) was logical due to the reciprocal trend in residence time with change in vertical hydraulic gradient. The reason for the negative correlation between $K_v$ and silt&clay percentage could be due to a lower percentage of clay and silt in the vertically sampled sediment core (from the river-bed) having a sediment pore clogging effect, and therefore exerting a limiting influence on ɧ. Moreover, clay and silt exhibit distinctly different behaviour during downward and upward flux: in downwelling flow, the small particles fill the sediment



pore spaces; in upwelling flow, the small particles move out of pore spaces. These differences

explain the variation of $K_v$ and porosity in river-bed sediments (Datry et al., 2015).

**4.3 Quantifying the magnitude of VHE and residence time**

The magnitude of radon activity ($BqL^{-1}$) and diel temperature variation at the different sediment

depths were measured to assess the magnitudes of temperature-derived and radon-derived VHE

(VHE ($VHE_t$ and $VHE_{Rn}$, respectively), and the associated residence times ($tr_T$ and $tr_{Rn}$) in the

four riffle-pool areas of the meandering stream. The in-situ sampling procedure was conducted

at depths of 0 (surface water), 5, 25, 45, 65, 85, and 105 cm below the sediment bed.

Based on the temperature and radon activity variations in different sediment layers, the

magnitude of VHE was estimated at the 33 testing points located parallel and perpendicular to

the riffle-pool areas (Figure 1A). The one-dimensional heat transport model (Equation 1) was

used to estimate the temperature-derived $VHE_t$ (Table 2). The positive and negative $VHE_t$

values reflected downwelling and upwelling water movement, respectively (Hyun et al., 2011).

The $VHE_t$ values ranged from -1082mm/day to 1803 mm/day, indicating considerable variation

due to the complex geomorphological conditions of the thalweg paths (Kasahara and Wondzell,

2003). The temperature-derived and radon-derived VHE ($VHE_T$ and $VHE_{Rn}$) at the 33

sampling points are illustrated in Figure 4.

The values of $VHE_t$ along the riverbed sediments of each zone reflected the significant

subsurface lateral flow, which is dominated by downward flow in the pools and upward

movement in the riffle segments. The highest $VHE_t$ values (Figure 4) occurred in B18 (1803

mm/day), B7 (1044 mm/day), B22 (1008 mm/day), B6 (986 mm/day), B23 (-1082 mm/day),

B10 (-1050 mm/day), and B8 (-890mm/day); these locations contained the lowest amounts of

silt and clay. This was in agreement with results reported by Kennedy et al. (2009), where high

$VHE_t$ values resulted from the abundance of sand and gravel content in river bed sediments





and erosional zones along the thalweg paths, and high $K_v$ values. Furthermore, the downwelling patterns were mostly observed at the river boundaries (river bank), such as B1, B7, B16 and B22 (Figure 2), whereas the testing points located at the central sections of the river and with higher elevation, such as B2, B10, B17, B23 and B26, exhibited upward exfiltration of water into the stream. The downward infiltration is driven by river sinuosity and the upward movement is mainly caused by the higher groundwater level (during winter), in this region (Cardenas, 2009). Stream sinuosity areas are considered as kinematic zones influenced by gradient pressure between downstream and upstream along river bends (Boano et al., 2014), where gradient-driven hyporheic flux forms the local flow paths. These complex flow paths indicate the effect of regional and local hyporheic flow patterns which play a significant role in regulation of the biogeochemical processes (Gomez-Velez et al., 2015).

To accurately estimate VHE, in situ radon activities were measured. The stream had a mean radon activity of 0.61±0.13 Bq/L, based on 41 surface water samples collected between 11 am and 6 pm on 8 February, 2015). The radon activity showed spatial variation, with a mean value for the surface water samples of 0.38±0.098 Bq/L and 0.93±0.28Bq/L in the riffle and pool cross sections, respectively. The main reason for the discrepancies in stream radon activity could be attributed to the change in stream water gas transfer velocity due to changes in the river depth and width across the riffle-pool sections (Cranswick et al., 2014). For example, in Z1 the mean river depth and width were 0.89 m and 213 m, respectively, but in Z4 these values were 2.05 m and 74.4 m, respectively. Also, the probable discharge of groundwater in riffle segments was another likely cause of disparity in stream radon values, including the negative values (Figure 4). The increase in surface water radon activity was strengthened by upward gradient, which was mostly observed across the riffle sections. The mean radon activities of the alluvial aquifer at the deepest sampling point (105 cm below the river-bed) ranged from



2.02Bq/L (B7) to 8.19 Bq/L (B30) with a mean standard deviation of 4.58±1.84 Bq/L (Figure 5).

Considering the results presented in Figure 4, it was evident that at testing points where groundwater exfiltration occurred, $VHE_{Rn}>VHE_T$, while at downwelling points such as B1 or B18, the VHE was overestimated by temperature ($VHE_{Rn}<VHE_T$). These discrepancies could possibly be ascribed to sub-surface lateral flow, the degree of aquifer-sediment heterogeneity, and the potential thermal transport heterogeneity associated with the sediments (Irvine et al., 2015; Rau et al., 2012). Furthermore, at some testing points such as B7, B10 and B18, the difference between $VHE_T$ and $VHE_{Rn}$ was significant. To further investigate these discrepancies, the vertical variations in diel temperature, radon activity and EC at 5, 25, 45, 65, 85 and 105 cm depths below the sediment bed) were measured (Figure 6 and Figure 7).

For profiles B3, B11, B13, and B19, the EC variation was relatively constant at different depths. There was increased diel temperature variation and a slight increase in the radon activity of the sediments, which was due to infiltration of groundwater. In the profiles with low downwelling/upwelling rates (e.g. B2, B10, B20, Figure 5), as depth increased there was increased EC, but decreased radon activity and temperature. In locations with high rates of upward flux (B29 and B32), the temperature envelopes represented constant temperature at different depths, but dramatically increased radon and EC values.

Radon activity was augmented at mid-depths of profiles such as B1, B10, B15, B23, B26, B29 and B32, and in some cases these values were higher than the radon activity reported by Meghdadi and Eyvazi (2017) for alluvial aquifers. This could be due to heterogeneity in the sediment radon production rate arising from variation in metal oxide radium adsorption-desorption, which mostly occurs at redox boundaries, or the sediment mineralogical heterogeneity (Lamontagne et al., 2011). Therefore, the sudden increase in radon activity at





mid-depths was due to geochemical processes occurring within the hyporheic zone or the sediment mineralogical heterogeneity.

Profiles B2, B3, B15, B17, B20, and B26 appeared to have downwelling flow at shallow depths of up to 0.45 m below the riverbed, because of the large diel temperature variations and low radon and EC activities. However, at higher profundities (>45 cm), upward flux was indicated by the constant temperature and sudden increase in radon activity and EC, except for at B3. These opposing flow patterns occurred mostly at depths greater than 45 cm, except for at B17 and B32 where this occurred at 25 cm, providing evidence that hyporheic flux is a part of a larger regional flow field. The high magnitudes of EC and radon at the greatest sampling depth (105 cm) at B10, B13, B26, B29, B30 and B32 were a result of interaction between groundwater from shallow aquifers with hyporheic water, mostly in areas that were close to alluvial stream channels. On the other hand, the relatively constant values of radon and EC in upwelling profiles such as B20, B17 and B22 did not display the elevated radon activity at the deepest testing points, which indicated that upward fluxes originated from the hyporheic shallow water instead of larger groundwater sources.

The general assessment of flow direction, considering all the information from positive and negative VHE values, and EC, temperature and radon profiles, was as follows: in the pool segments the hyporheic flow was downward then switched to horizontal flow, then finally groundwater ex-filtration occurred in riffles. Similar trends have been reported by Cranswick et al. (2014) and Cook et al. (2004).

### 4.4 Simulation of spatial distribution of VHE using the PHAST simulator

Using radon activity and temperature to quantify the magnitude of VHE in thalweg paths characterised by riffle-pool sequence morphology is confounded by the complex subsurface patterns. These arise from heterogeneity in the sediment mineralogical properties, hyporheic





water interaction with regional groundwater flow, and geochemical interactions between sediment and hyporheic flow see section 4.3). To address this difficulty it was decided to apply simulation models (Käser et al., 2014).

Accordingly, the spatial distribution of VHE was simulated using PHAST modelling, not only to achieve conceptual insights relating to the spatial variations of VHE in the meandering channels, but also to assess the operational accuracy of the simulator against temperature and radon to quantify VHE in the thalweg paths containing multiple riffle-pool sequences. The model properties following set up and calibration procedures are briefly described in the following sections.

After employing the input files (for more information see Parkhurst et al. (2010) and Winston, 2009), the model was run more than 3000 times to simulate hyporheic exchange. The performance of the model in estimating the vertical flow direction (VFD) in each zone was tested using a plot of residuals vs vertical head difference (VHD), which is illustrated in Figure 8A. The ratio of the absolute values of residuals ($\varepsilon$) to the absolute values of VHD were applied to assess the reliability of the results. If the ratio of $\left|\varepsilon\right|/\left|\text{VHD}\right|$ is less than one, this indicates that the model performance is reliable (Käser et al., 2014). The vast majority of simulated VHD values were larger than the associated residuals in each of the four zones (Figure 8A), which indicated that the model more correctly characterised the hyporheic flux. Therefore, the model was considered to be a valid simulator of VHD and VHE.

However, although the model provided a reliable prediction of the hyporheic flow patterns, in Z3 and Z4 the model slightly overestimated the VHD, especially in downwelling segments, and underestimated the upwelling heads. This discrepancy in the model performance may have been caused by an overly significant influence of the limited downwelling points on the model calibration (Stewardson et al., 2016; Wondzell et al., 2009). Because the model operates on the





basis of equal weighting to each observation, overestimation of the head gradient may have

occurred in the model set up. The model was not initially designed for heterogeneous basal

flux, so that in the model the linear variation of basal flux conducted among the lateral

boundaries would lead to increased hydraulic gradient on one side and decreased on the other

side.

The reliability of the model calibration was investigated in each zone separately considering

the magnitude of the root mean square error (RSME) and mean absolute error, as follows:

$$RMSE = \sqrt{\frac{1}{n} \sum_{i=1}^{n} (h_{sim,i} - h_{obs,i})^2} \qquad (7)$$

$$MAE = \frac{1}{n} \sum_{i=1}^{n} h_{sim,i} - h_{obs,i} \qquad (8)$$

In equations 7 and 8, n refers to the total number of measurements, $h_{sim,i}$ refers to the simulated

head at point i, and $h_{obs,\,i}$ refers to the observed head. The values of the observed head were

defined in the model based on interpolation of the in situ head measurements and the Digital

Elevation Model input file of the stream-bed sediments. The results of the error analysis

indicated that the model best fit was at Z4 (RSME=0.07 and MAE=0.13), followed by Z1

(RSME=0.09 and MAE=0.21), Z2 (RSME=0.1 and MAE=0.18), and Z3 (RSME=0.11 and

MAE= 0.25).

The spatial distribution of the VHE flux is illustrated in Figure 8B. The positions and spatial

distributions of the downwelling and upwelling segments were similar to the results obtained

from temperature and radon analysis. In most of the pool sections, especially in Z2 and Z4, the

spatial distributions of the upwelling and downwelling locations were similar to the chosen

testing points (Figure 8B). In Z1 and Z3, except for downstream of riffles, the results were

quite similar to those from the radon and temperature analyses. The 7 m × 7 m mesh

discretisation values were employed during the simulation procedure to provide sufficient





details of the flux spatial variations. Choosing the finer mesh size would enable more detailed flow patterns, but use of the finer mesh was not possible in this study due to the massive size of the study zones and limitations of the computer hardware to run the model.

The values of VHE obtained from temperature, radon activity and the simulation are compared in Figure 8C. It was evident that the magnitudes of VHE predicted by model were more correlated with temperature-derived VHE ($R^2_T$= 0.96 while $R^2_{Rn}$=0.76, Figure 8C). This greater correlation of the simulated results with the temperature-derived results was due to the lower sensitivity of temperature to subsurface geochemical reactions and heterogeneity in the sediment mineralogical properties compared with radon.

## 4.5 Temperature-derived residence time vs radon-derived residence time

The values of $tr_T$ and $tr_{Rn}$ and their associated error bars (5[th] and 95[th] percentile) are illustrated in Figure 9A. The magnitude of $tr_T$ ranged from 0.9 at B23 to 5 in B16, with a mean value of 1.87±1.26, while $tr_{Rn}$ varied from 0.6 at B2 to 4.1 at B16, with a mean value of 2.11±1.17. A scattered relationship rather than a systematic relationship was observed between $tr_T$ and $tr_{Rn}$, as well as an evident discrepancy between $tr_T$ and $tr_{Rn}$ at points with high and low $K_v$. For example, at points with low $K_v$ values such as B16 ($K_v$ = 0.503 m/day) and B12 ($K_v$ =0.62 m/day), the magnitude of $tr_T$>$tr_{Rn}$: at B16, 5 and 4.1 days were observed for $tr_T$ and $tr_{Rn}$, respectively and at B12, these values were 3.4 and 2.1 days, respectively. In contrast, at testing points where a high $K_v$ magnitude, such as B23 ($K_v$=3.41 m/day), the $tr_{Rn}$ was higher than $tr_T$. This finding confirmed that obtained by Ferguson and Bense (2011). Schornberg et al. (2010) provided evidence that the proximity of two contrasting hydraulic conductivity areas leads to heat-derived fluxes that do not reflect the water advective flux, especially near the boundaries of the two zones with opposing $K_v$ values.



The values of $tr_T$ at the downwelling points (e.g. B1, B7, B16, B19, B22, B27, B30 and B31) appeared to be shorter than $tr_{Rn}$, and at the upwelling points (B2, B6, B15, B20, B23, B26, B29 and B32), $tr_T$ was longer than $tr_{Rn}$. This may suggest that in the zones where upward movements are dominant, the sediments were mostly composed of material with low $K_v$ capacity and a

small proportion of the sediments consisted of high $K_v$ material, such as sand and gravel.

The measured values of $tr_T$ and $tr_{Rn}$ and the upper and lower uncertainty boundaries at the 95 percent confidence interval (based on the Monte Carlo analysis) are illustrated in Figures 9B and 9C. The values of $A_0$ and $A_e$ (from equation 5) varied randomly and followed a normal distribution, based on the mean and standard deviation of the calculated values. However, at a

few points (B13, B16, B17, B18 and B31), the estimated radon-derived residence time exceeded the upper and lower uncertainty boundaries, but the value of $tr_T$ lay well within the uncertainty boundaries. Because subsurface temperature is more influenced by adjacent flux than is radon activity, the temperature-derived residence time (and flux) had less variation at the different testing points than $tr_{Rn}$, and therefore is proposed to be more representative than

radon-derived flux and residence time.

The finding that values of $tr_T$ were shorter than $tr_{Rn}$ might be due to river water sampling of the radon activity at points with groundwater discharge. At these locations, radon activity is underestimated and subsequently the residence time is overestimated, and therefore appears to indicate longer residence time compared with temperature. The values of $tr_{Rn}$ were 1.12 orders

of magnitude higher than $tr_T$, which could be due to the proximity of the testing points to the area where the $K_v$ is low. Ferguson and Bense (2011) have revealed that temperature-derived flux (and residence time) can provide up to two orders of magnitude variation from the actual advective flux when there are up to three orders of magnitude change in the values of $K_v$ in adjacent zones. Therefore, differences between $tr_T$ and $tr_{Rn}$ can be less than the differences

between $tr_T$ and true advective dispersion flux.



## 5. Conclusion

This study quantified the VHE and residence time using diel temperature variations and radon activity at different depths of the Ghezel-Ozan river sediment bed. The hydraulic conductivity of the sediment bed materials was also used, in the four divided zones of the study watershed, to characterise the morphological features and their influence on the VHE pattern. At testing points with relatively lower altitude, downward flux occurred and the values of $K_v$ were lower compared with points at higher elevations where there was upward flux. This was affected by fine sediment particles which clogged the pore spaces of the stream bed sediment during downward movement. The values of diel temperature, radon activity and EC variations at the different depths revealed the highly dynamic characteristics of the subsurface flow, while the hyporheic exchange was mainly effected by larger scale regional subsurface flow.

Also, in this study, the results obtained from the PHAST model simulator were compared with the values of $VHE_T$ and $VHE_{Rn}$. The error analysis of the model output results indicated the highly acceptable performance of the PHAST model in estimating hyporheic vertical flux, due to the low value of $|\varepsilon|/|VHD|$. Furthermore, the spatial distribution map of downwelling and upwelling areas, as well as the magnitude of the hyporheic flux, was described by the PHAST model: downward flux was mainly observed in the central river sections, and upward flow mainly occurred along the river boundaries. The close correlation especially between $VHE_{PHAST}$ and $VHE_T$ ($R^2 > 0.95$) indicated the high suitability of PHAST model to characterise hyporheic vertical flux.

The results of this study demonstrate that the meandering reaches can be considered as kinematic areas, as the hyporheic vertical flux in river curvatures that possess multiple sequences of riffle-pool points is extremely dynamic, and influenced by the river morphology and the pressure gradient between upstream and downstream.





**Acknowledgment**

This study was financially funded by Zanjan's Natural Resources Office, Division of Water Resources Management (Grant Number: ZNWRM/21248-810/A2). I would like to express my gratitude to all of the laboratory staff, especially Mr Kambiz Amini from the Division of Water Resource Management (Tarom region Branch) for their special assistance during sampling procedures.

**Figures Caption**

Figure1. Location of the study watershed (A); schematic diagram of the piezometer for calculation of $K_v$ and MLS for measurement of the streambed temperature at different depths.

Figure 2. Interpolated elevational contour map of the river bed sediments at the four study zones.

Figure 3. The principle component analysis between $K_v$ and sediment hydraulic and physical properties.

Figure 4. The magnitudes of temperature-derived VHE ($VHE_T$) and radon-derived VHE ($VHE_{Rn}$) at each testing point; negative values refer to the hyporheic upward flux and the positive values indicate downward flux in the hyporheic zone.

Figure 5. The magnitude of sediment bed radon activity (measured in situ at 1.05 m sediment depth) and the associated calculated residence time.

Figure 6. Diel temperature, radon activity and EC variations at different depths for Z1 and Z2; there are discrepancies between radon-derived and temperature-derived VHE.

Figure 7. Diel temperature, EC, and radon variations at different depths for Z3 and Z4 at the sampling points with the greatest discrepancy between $VHE_T$ and $VHE_{Rn}$.

Figure 8. (A) Testing the PHAST model performance using plot of simulated vertical head residuals vs observed vertical head difference for the entire watershed, on the basis of running the calibrated model; the cyan area indicates where the error is higher than simulated results. (B) The summary of RSME and MAE for the four study zones, the spatial distribution of downwelling and upwelling locations, and the modelled $VHE_{PHAST}$ ranges for each zone. (C) Comparison between $VHE_{PHAST}$ with $VHE_T$ and $VHE_{Rn}$ and the subsequent magnitude of correlation coefficients ($R^2$).

Figure 9. (A) Relationship between $tr_{Rn}$ and $tr_T$; error bars indicate the 5th and 95th percentiles. (B) Radon-derived residence time ($tr_{Rn}$). (C) Temperature-derived residence time ($tr_T$). The lower and upper uncertainty boundaries at 95% confidence interval were obtained by Monte Carlo analysis.

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

**Tables**

Table 1. The stream bed sediment grain size distribution at the four study zones

|  | Z1 | Z2 | Z3 | Z4 |
|---|---|---|---|---|
| <0.075 mm cumulative weight | 23.2 | 26.5 | 17.8 | 19.4 |
| <2 mm cumulative weight | 75.6 | 74.2 | 81.7 | 80.4 |
| $D_{50}$ | 0.94 | 0.87 | 1.21 | 1.1 |
| Coefficient of uniformity (ʄ) | 1.84 | 2.87 | 2.04 | 2.41 |
| Porosity | 0.43 | 0.40 | 0.42 | 0.41 |

Table 2. Statistical analysis of VHET and VHERn and stream sediment bed vertical hydraulic

conductivity. The negative and positive VHE values refer to upward and downward flux,

respectively, the magnitudes of the mean and standard deviation were calculated based on the

absolute values of VHET and VHERn.

| Study segment | Test point | Parameter | Range | Standard deviation | Coefficient of variation | Mean |
|---|---|---|---|---|---|---|
| Z1 | B1 to B12 | Kv (m/day) | 0.49 to 2.82 | 0.739 | 0.511 | 1.445 |
|  |  | VHE Rn (m/day) | 1.22 to -1.32 | 0.368 | 0.554 | 0.664 |
|  |  | VHE T (m/day) | 1.04 to -1.05 | 0.294 | 0.442 | 0.665 |
| Z2 | B13 to B21 | Kv (m/day) | 0.27 to 2.07 | 0.614 | 0.560 | 1.096 |
|  |  | VHE Rn (m/day) | 0.26 to-1.55 | 0.555 | 1.226 | 0.680 |
|  |  | VHE T (m/day) | 1.8 to -.98 | 0.506 | 0.730 | 0.693 |
| Z3 | B22 to B27 | Kv (m/day) | 0.64 to 3.76 | 1.333 | 0.699 | 1.907 |
|  |  | VHE Rn (m/day) | 0.55 to -.9 | 0.216 | 0.372 | 0.582 |
|  |  | VHE T (m/day) | 1.04 to -1.08 | 0.234 | 0.305 | 0.767 |
| Z4 | B28 to B33 | Kv (m/day) | 0.61 to 2.425 | 0.648 | 0.456 | 1.421 |
|  |  | VHE Rn (m/day) | 0.64 to -1.15 | 0.294 | 0.453 | 0.647 |
|  |  | VHE T (m/day) | 1.08 to -0.78 | 0.199 | 0.270 | 0.736 |



Figure1. Location of the study watershed (A); schematic diagram of the piezometer for calculation of Kv and MLS for measurement of the streambed temperature at different depths (the scale bar at the bottom of the figure refers to Z1 to Z4).



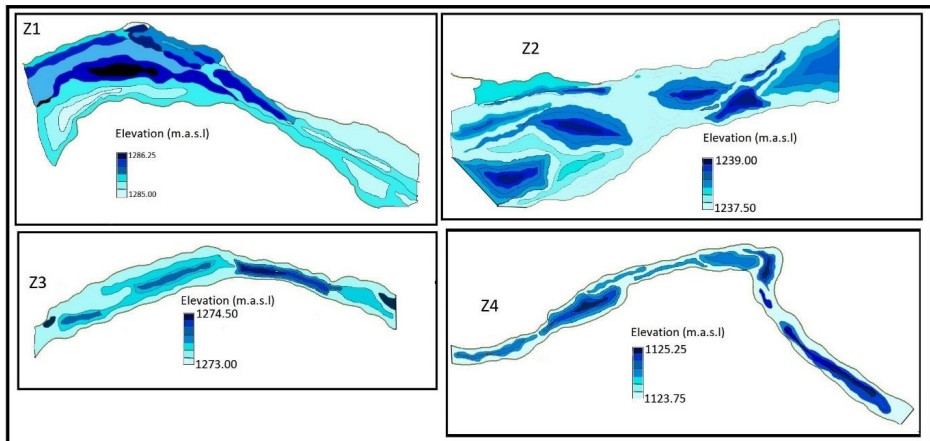

5    Figure 2. Interpolated elevational contour map of the river bed sediments at the four study zones.

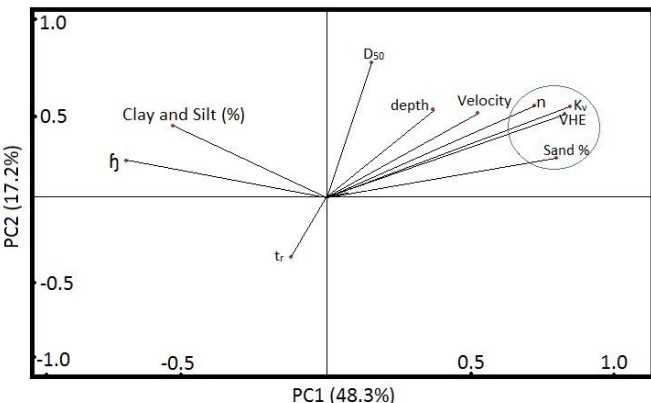

Figure 3. The principle component analysis between $K_v$ and sediment hydraulic and physical properties.



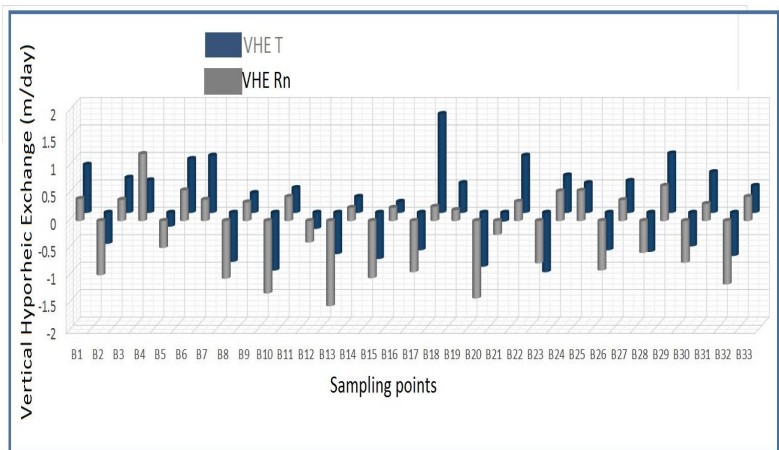

Figure 4. The magnitudes of temperature-derived VHE (VHE$_T$) and radon-derived VHE (VHE$_{Rn}$) at each testing point; negative values refer to the hyporheic upward flux and the positive values indicate downward flux in the hyporheic zone.

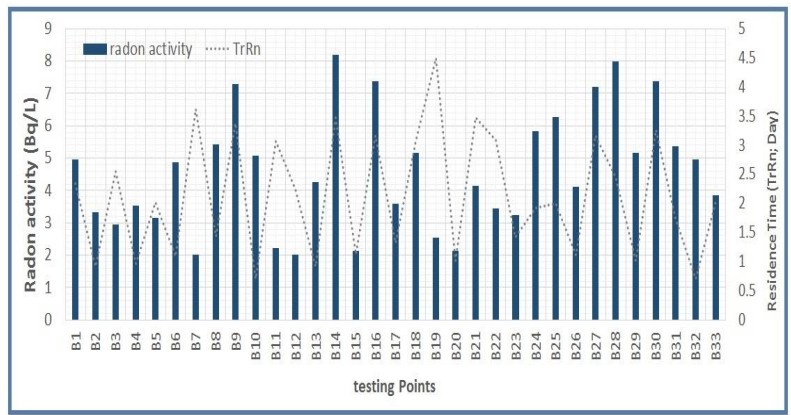

Figure 5. The magnitude of sediment bed radon activity (measured in situ at 1.05 m sediment depth) and the associated calculated residence time.





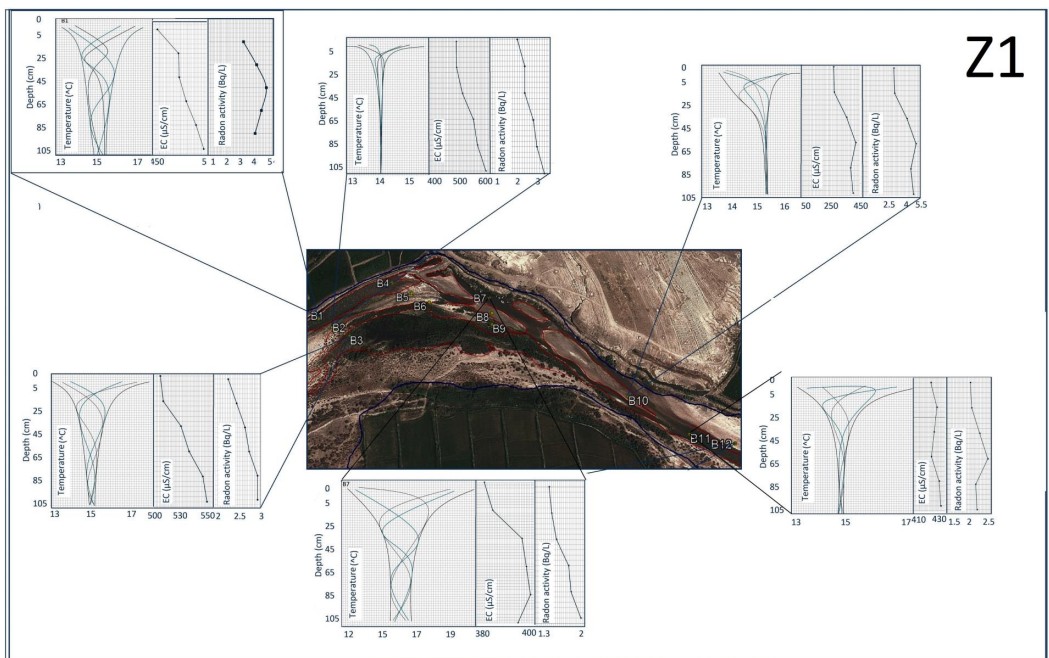

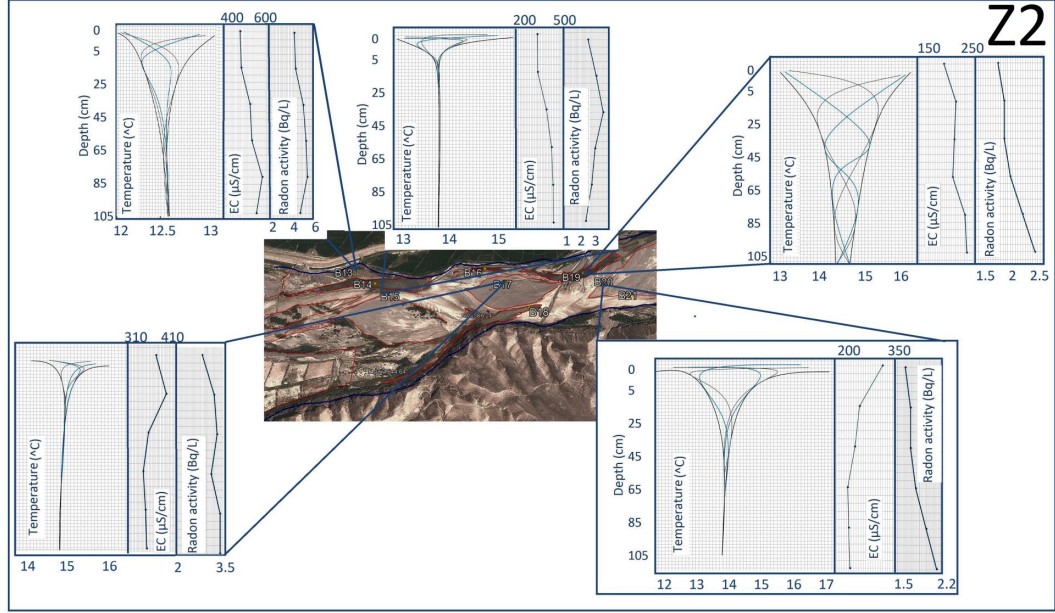

Figure 6. Diel temperature, radon activity and EC variations at different depths for Z1 and Z2; there are discrepancies between radon-derived and temperature-derived VHE.







Figure 7. Diel temperature, EC, and radon variations at different depths for Z3 and Z4 at the sampling points with the greatest discrepancy between VHE$_T$ and VHE$_{Rn}$.



Figure 8. (A) Testing the PHAST model performance using plot of simulated vertical head residuals *vs* observed vertical head difference for the entire watershed, on the basis of running the calibrated model; the cyan area indicates where the error is higher than simulated results. (B) The summary of RSME and MAE for the four study zones, the spatial distribution of downwelling and upwelling locations, and the modelled $VHE_{PHAST}$ ranges for each zone. (C) Comparison between $VHE_{PHAST}$ with $VHE_T$ and $VHE_{Rn}$ and the subsequent magnitude of correlation coefficients ($R^2$)

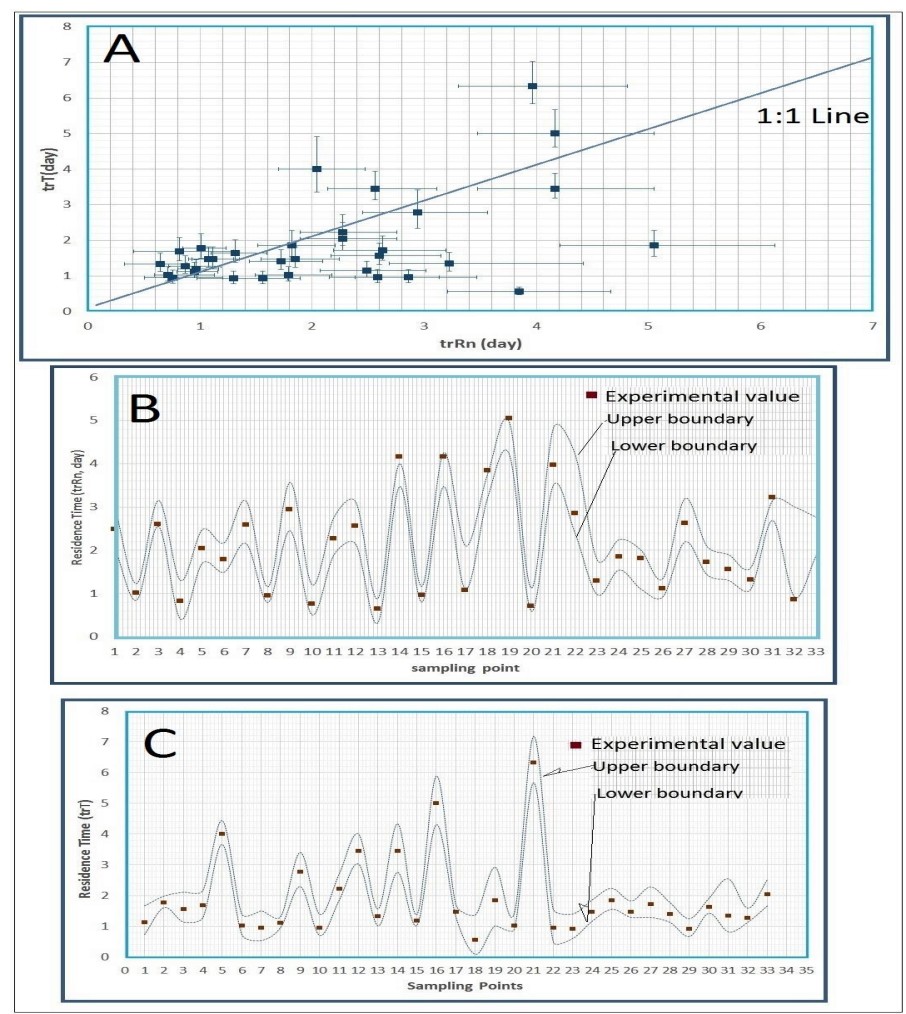

Figure 9. (A) Relationship between $tr_{Rn}$ and $tr_T$; error bars indicate the 5th and 95th percentiles. (B) Radon-derived residence time ($tr_{Rn}$). (C)

Temperature-derived residence time ($tr_T$). The lower and upper uncertainty boundaries at 95% confidence interval were obtained by Monte Carlo

analysis.

