# Peer review of "Quantifying Vertical Hyporheic Exchange and hyporheic residence time in thalweg paths of meandering streams characterized by multiple riffle-pool sequences morphology"

_Hydrology and Earth System Sciences, 2019_

## Short Comment (SC1) · 14 Oct 2019

Hello,

can you explain in more detail which temperature information you used to calculate the vertical exchange rates? Applying the simple steady state approximation to calculate exchange fluxes is tricky, because not all input data is acceptable. How long did you measure temperatures? The temperature envelopes in one of your figures suggest that you measured longer time series?

Best regards, Christian

---

## Referee Comment (RC1) · Anonymous Referee #1 · 1 Nov 2019

The manuscript submitted by Meghdadi et al. deals with the application of natural tracer and temperature to quantify hyporheic exchange fluxes for riffle-pool systems. Although the overall topic is very interesting and the presented data shows some good potential, the manuscript suffers from some major flaws. Especially how the data was used to quantify hyporheic exchange is highly questionable and in my opinion not acceptable for publication. In addition the manuscript in some parts is poorly structured and the presented figures are very confusing and of poor quality. Therefore I cannot recommend the presented manuscript for publication in HESS. Major Comments:

[Figure]

1) Temperature method The authors used vertical temperature profiles to quantify hyporheic exchange based on a steady state assumption (Equation 2). At the same time data (diurnal temperature fluctuation) is presented (Figure 6) clearly showing that this assumption is violated. Here I think the unusual high exchange fluxes >1 m/d may result from the violation of steady state assumptions. 2) Radon method Beside temperature the authors used vertical Radon profiles to quantify hyporheic exchange. Based on the presented equations (Equation 5) and the presented data it is a very unclear to me how to derive upward fluxes using Radon. Under strictly 1D assumptions upward fluxes require a Radon profiles where the Rn activity is increasing towards the streambed interface (young water in the depth and old water at the streambed interface). For example for site B7 the Rn profile indicates in Figure 6 indicates that the water age is increasing in depth. Under 1D assumption this indicates infiltrating conditions and a negative flux, however the authors present an upward flux. And almost all of the presented Radon profiles look like this. Also how did the authors distinguish between water from the hyporheic zone and groundwater inflow? 2) 1 D Assumptions Hyporheic flow fields are known to be quiet complex and three dimensional in nature. Hyporheic flow is influenced by 1) the hydrodynamical conditions in the open channel flow, 2) scale depended characteristics of the streambed and 3) interactions with the regional groundwater flow. Further hyporheic flow in general is known to be three dimensional in nature. Many studies that deal with the hyporheic zone do use 1D assumptions but mostly this is used to estimate fluxes for shallow areas of the streambed (<10 cm). The authors evaluated temperature and Radon profiles taken from 1m deep wells using 1D assumption. At this depth lateral flow components can't be neglected as also lateral groundwater flow might be important at this depth (which is also mentioned by the authors).

3) Numerical model There are many modelling studies available that are trying to represent hyporheic fluxes accurately as a function of the morphological conditions of the streambed. Almost all these modelling studies mention that, beside an accurate representation of the streambed topography, the pressure variations along the streambed

are necessary to predict hyporheic exchange patterns. The numerical approach, which is only poorly described in the methods chapter, represents river flow in a very simplified manner by applying source terms. Does that mean that the authors need to explicitly specify whether the stream loses or receives water from the subsurface? If this is the case the whole simulation with the purpose to predict in- and exfiltration areas and associated fluxes does not make sense as it is already pre-defined in the model setup. Minor comments Please add consecutive line numbers. It is hard to address issues in the manuscript without them. Page 2 line 22: Explain meaning of kinematic zones. Page 3 line 3: Hydrodynamic conditions in the stream are also known to be important for hyporheic exchange. Page 4 line 17: I would not describe a 10 years old study as a recently published study. Page 4 line 19: replace "In other study,..." with "In another study,.." Page 5 line 1: remove "recently" Page 5 line 4: The whole paragraph fits better into the methods chapter Page 5 line 4: Remove "recently" Page 5 line 4: What do you mean with "...it employs a set of differential equations..." Page 5 line 23: What do you mean with "apply the vertical variations of diel temperature..." Page 6 line 11: What is a "dominant geochemical unit"? Page 7 line 21: replace "calculated" with "measured" Equation 1: I know this equation only using $v=q/n$ (n=porosity) as the advective velocity. And please add SI units for the different components Equtaion 2: This is a steady state solution and is not directly result from equation 1. Please add made assumptions. In general the entire methods chapter needs revising as much more detail has to be provided. Equation 4: please add units (for all equations). Page 10 line 15: What is "ID" Page 10 line 15: I do not understand the meaning for the sentence beginning with "As an assessment ….." Page 11 line 1: Add "The" at the beginning of the sentence. Equation 5: Units Page 11 line 21: Was this equation ever used before in a similar fashion? PHAST modelling section: The whole chapter needs much more information 1) about the code 2) the numerical implementation, 3) how river flow is presented and 4) the applied boundary conditions specifically the lower boundary. Is it no flow or is there some kind of groundwater inflow? Also a much better explanation is needed why and how the authors "transformed the river boundary conditions into

source terms". Results and Discussion: Usually I prefer that Results and Discussion are two separate chapters. Page 13 line 11: What do you mean with "..33 testing points along the riffle-pool sequence of the meandering river where identified". Also the entire paragraph better should be shifted into the methods chapter. Table 1: what does the 10 mean in column Z4 Page 16 line 7: I think Cardenas never worked on a river in Iran. Page 16 line 12: This sentence sounds like that the authors assume that Radon is the reference method obtaining accurate estimates for hyporheic fluxes. Page 16 line 17: Groundwater inflow also does influence the in-stream Rn activity. Page 19 line 2: Which difficulties? Page 19 line 6: "..to asses the operational accuracy of the simulator against temperature and radon.." This statement implies that estimates from the temperature and radon method are accurate. Page 19 line 10: The whole paragraph should be shifted to the methods chapter. Figure 1: The whole figure is very confusing with too much sub-plots. Figure 2: Why are the plots all differently sized? This is kind of sloppy. Also remove the frames Figure 4: This Figure is also hard to read please present differently Figure 5: dee above Figure 6: Profile plots are too small. Maybe less profiles are the better option Figure 7: see above Figure 8: see above Figure 9: Again very sloppy. Each of the plots are differently sized.

---

## Referee Comment (RC2) · Martijn Westhoff (Referee) · 11 Dec 2019

Dear authors,

Because one reviewer missed the deadline and did not react to any of my e-mails I decided to review the manuscripts myself. Since I am also editor of this manuscript, I would like to stress that I did not read the first review when doing my own. So my decision as editor (taking the other review into account as well) may differ from that as reviewer.

The manuscript describes different methods to quantify the hyporheic exchange fluxes and residence times on several locations along a river characterised by pool-riffle sequences. The different methods were 1) using vertical temperature profiles; 2) using vertical radon profiles; 3) using vertical EC profiles and 5) by calibrating a hydrological model on groundwater heads.

While the results of these different methods has the potential to gain insights on the pros and cons of the different methods as well as insight in how such systems work, the novelty of this work is not very clear to me. The four objectives are rather detailed and more at the level of sub-questions, while an overarching objective was missing.

Furthermore, the structure of the manuscript could be significantly improved. There is little structure, especially in the results and discussion section, without much synthesising of the results from the different methods. Among others: my advice would be to separate the results from the discussion section. Currently, it is often unclear what the finding of this manuscript are and what findings of others are.

Altogether, I find these shortcomings too much to warrant major revisions, but I encourage a resubmission as a new manuscript (if the authors can answer all raised questions satisfactory – especially the novelty of the manuscript should become clear). More detailed comments are given below:

P4 L17: this paragraph is merely namedropping. Indicate limitations/made assumptions etc. that makes the novelty/niche of this study clear.

P5 L4: This paragraph contains too much detail of the model. This should go to the methods section.

P5 L20: which of these aims is the main objective?

P9 Eq. 3: Why not simply stating that VHE equals q? You can make this statement already when describing Eq. 1

P10, section 3.2: More information should be given here: Describe that this test is

done with a slug test, where the sediment is inside the tube. Also refer to the original literature in which this method has been described (i.e Chen 2000 - Environmental Geology)

P10 L15: So Kv was about 8 times higher than ID? I guess it should be the other way around. Also explain why this is important: is it to say that the resistance of the tube is much lower than that of the soil?

P12 Eq. 6: a small sketch of all the fluxes in and out of the grid cell would be useful. Is there also lateral flow of groundwater and or river water?

P13 L12-13: What do you mean with this? What do you mean with flux momentum: density times velocity? Also add a unit after '0.5' (do this after all numbers throughout the document!)

P13 L17-18: This is a clear example of how your own results are mixed with those from literature. To me it is unclear what your own findings are and what comes from literature. By separating the results from the discussion, this can easily be avoided.

P14 L1-2: where do these results come from?

P14 L6: How are these Kv values derived?

P15 L1-2: This is another example where it is unclear if these are your own findings or findings from others

P15 L15: The reference to Kasahara and Wondzell (2003) seems a bit odd here, since you are talking about your own results.

P16 L5-6: the water fluxes are by definition driven by a difference in hydraulic head. The question is more why at one point the groundwater head is higher than the river head while at other points it is the other way around.

P16 L12: This implies that the temperature derived ones are not accurate.

P17 L5: how do you know that VHE_T is an overestimation? Maybe radon underestimated it

P19 L15: a ratio of one means that the error is as big as the head difference, right? That sounds like a large error to me.

P20 L14: How large is the RMSE and MAE in relation to the VHD? (also add units after the numbers)

P22 L1-2: Could it be that direct heating of the streambed by solar radiation plays a role here? This would mean that more heat is being transported in downward direction, leading to a lower t_r

P22 L3-5: Why would this argument not be valid for tr_Rn?

P22 L12-15: Why not the other way around? The influence by adjacent fluxes introduces, in my opinion, an error

Table 1: Within each zone more than one sample has been taken, right? The result of which sample is shown here? I suspect it is the average of all the samples in one zone. However, that does not say anything given the large heterogeneity within each zone

Fig. 1:

- indicate the insets in the middle figure with squares instead of circles (the same area of the zoomed displays)

- The North arrow seems to be wrong given the square drawn in the top right map

- The sampling points are hardly visible

- It seems from this figure that the installation given in panel B has only been installed at sampling point B7, but I understood that it has been installed at all sampling point. Is that correct

Fig. 2: add a scale bar

Fig. 4: It would be easier to interpret this figure when it is a 2D figure

References:

Chen, X. Environmental Geology (2000) 39: 1317. https://doi.org/10.1007/s002540000172

Kasahara, T., and Wondzell, S. M.: Geomorphic controls on hyporheic exchange flow in mountain streams. Water Resources Research, 39(1), SBH 3-1. doi:10.1029/2002WR001386
* * *

---

## Editor Comment (EC1) · Martijn Westhoff (Editor) · 13 Jul 2020

Dear authors,

You have received two very critical reviews, and after studying your author responses I came to the conclusion to reject the paper.

The main reasons for this are:

- The novelty of the paper has not been made clear enough (just the fact that the study

has been done in another country or region does not make it novel.

- Reviewer 1 indicated a shortcoming in the methods where upwelling water should have higher radon concentrations close to the streambed compared to deeper in the soil. In you rebuttal you implicitly imply that you don't have upwelling zones, but only exchange of surface and groundwater.

However, if surface water flows into the subsurface, it has to come out somewhere as well.

- Reviewer 1 correctly questions the assumptions of the poorly described numerical model. In your response you did not explain which assumptions have been made in the model. Instead you describe the differences between analytical and numerical solutions, but this was not the point Reviewer 1 made. Altogether, I consider the author responses not convincing enough to warrant a major revision. Nevertheless, I think you have a nice dataset with the potential to be published, albeit in a completely new submission and taking all comments into account.

With kind regards,

Martijn Westhoff
* * *